# Optimization of Peer-to-Peer Power Trading in a Microgrid with Distributed PV and Battery Energy Storage Systems

**Hui Huang, Shilin Nie \*, Jin Lin, Yuanyuan Wang and Jun Dong**

North China Electric Power University, Department of Economic Management, Beijing 102206, China; hh@ncepu.edu.cn (H.H.); 120192206099@ncepu.edu.cn (J.L.); 120192206002@ncepu.edu.cn (Y.W.); dongjun@ncepu.edu.cn (J.D.)

**\*** Correspondence: 1182206131@ncepu.edu.cn

**Abstract:** Integrating distributed generation (DG) into the main grid is a challenge for the safety and stability of the grid. The application of peer-to-peer (P2P) technology in microgrids with distributed generation is expected to facilitate increased self-consumption of distributed and renewable energy, and the rise of prosumers' monetary benefits. A P2P energy trading model in microgrids with photovoltaic (PV) distributed generation and battery energy storage systems (BESSs) is proposed in this paper. We additionally designed a P2P electricity trading mechanism based on coalition game theory. A simulation framework of this model is presented which assumed a local community with 30 households under comprehensive constraints encompassing a customer load profile, PV system, BESSs, market signals including feed-in tariffs, and retail prices. Firstly, individual customers can post orders (purchasing orders or selling orders) and exchange information in a P2P energy trading market. Secondly, the microgrid operator can validate the orders based on how to achieve the minimum overall energy consumption in microgrids and set reasonable real-time purchasing and selling prices for P2P energy transactions. Thirdly, the orders can be automatically conducted and completed at the designed optimal price. This mechanism can be a practical solution motivating individual customers to participate in P2P electricity trading, assist with electricity cost reduction, benefit from electricity supply increases, and help the grid operators to make the most economically and socially friendly decisions.

**Keywords:** photovoltaic distributed generation; battery energy storage systems (BESSs); peer-to-peer power trading; microgrid; game theory

## 1. Introduction

Deterioration of the global environment and the depletion of fossil fuel energy has led to growing attention being focused on distributed and renewable forms of generation, like solar energy and wind power. Within a global context of rapid development of solar photovoltaic (PV), Australia has the leading position in the penetration of PV panels for residential users [1]. Globally, due to the shift of energy supply technology and patterns, energy markets are also undergoing a shift toward a decentralized and digital economy, while peer-to-peer (P2P) energy trading for distributed generation (DG) is finding its way into the energy sector. The application of P2P makes it possible for individual consumers to become prosumers and to share their excess energy with neighbors [2–4]. They can achieve a win–win situation by looking for a reasonable trading price and by making a deal in a seamless way [5,6]. P2P energy trading empowers customers to trade electricity at a P2P marginal price that is cheaper than the time of use (TOU) price and higher than feed-in tariffs (FIT),

respectively, which provides attractive savings for buyers and profit for sellers. In particular, it will support local consumption of distributed energy while reducing fossil energy consumption and carbon dioxide emissions.

In 2017, the Chinese government issued a guideline named A Pilot Approach to Facilitate the Construction of the Grid-Connected Microgrid [7]. Its conclusion can be summarized into the following four statements: (1) Generally, the maximum annual exchange capacity between a microgrid and the main grid is 50% of the total amount of annual electricity consumption in microgrids. (2) The microgrid operator (or associated agency) is responsible for the operation and maintenance management of the microgrid, the internal and external power balance, and electricity exchanges. (3) It is important to encourage the establishment of a price system for self-negotiation between any two parties where at least one uses a microgrid, as well as a trading mechanism for energy markets that covers various energy resources like cold energy, hot energy, and electricity. (4) Complying with market rule, the microgrid operator needs to take charge of electricity exchanges and the corresponding transmission and distribution costs. As a result, P2P as a new mechanism allows energy from one prosumer to be directly sold to another prosumer within the network at a negotiated price without any influence of a provincial transaction center. At the same time, the network served as a whole can sign a contract with the provincial transaction center for any external generation that is produced in the wholesale market. The microgrid operator is in charge of maintaining the distributed energy balance and managing consumption rather than the main grid operators, whose job it is to check the exchange capacity of the also controllable tie line. In addition, pressure to approve intermediaries' cost can be eased by including distribution costs within the microgrid into P2P electricity prices. Hence, P2P trading provides much promise both for academic circles and for the energy ecosystem.

The state-of-the-art literature relating to P2P trading in the energy sector is discussed in this paper and they can be classified into two main categories: (a) applications of P2P trading in the energy sector and (b) a pricing mechanism for a P2P energy trading market. In the first category, P2P energy transactions require advanced technical support, which is usually obtained by using online services based on information and communication technology [8]. A hierarchical system architecture has been proposed to identify and classify the key elements and technologies involved in P2P energy trading. In addition, a P2P energy trading platform has been presented and illustrated in a simulation whose results show that P2P energy trading can help with network and congestion management and allow more DG to be more widely shared within a community [9]. A two-level aggregation control technology for P2P energy trading in microgrids in communities was devised, which empowers users to follow their orders through a third party, i.e., an energy sharing coordinator. In the first phase, a constrained nonlinear programming (CNLP) optimization with a rolling horizon was used to minimize the community's energy costs. In the second phase, the control setting point was updated with a change of real-time measurement results, and was operated based on a price mechanism. It was found that a P2P energy trading model helps households to obtain 30% lower energy costs than the traditional energy trading model [10]. An architecture model for the design and interoperability of P2P energy trading components in microgrids was also proposed and simulated based on game theory, involving a specific customer-to-customer business model being introduced into a reference grid-connected microgrid and a core component of the bidding system, Elecbay, being presented [11]. Ottesen et al. proposed an aggregator exclusive available to the prosumer market, which would be intelligent enough to make electricity trading decisions on behalf of prosumers. The flexibility and stochastic planning of bidding in the day-ahead market with the help of an aggregator was investigated and the results showed that the system flexibility increases with the presence of an aggregator [12].

In the second category of the state-of-the-art literature, a novel dynamic pricing method was proposed to promote market-oriented decentralized energy transactions and provide the most economical benefits for owners with distributed generation [13]. A P2P energy trading mechanism for energy auctions was presented, which ensures the fairness and efficiency of energy auction by establishing a market design mechanism for energy auction. The Bayesian game strategy is used to

develop an optimal bidding strategy for distributed energy owners, so that each participant can obtain an efficient and economic bidding price. The results show that this model can maximize the utility of typical distribution network users [14,15]. Tushar W et al. introduced how motivational psychology encourages distributed energy owners to actively participate in P2P energy trading, and proposed a game theory P2P energy trading scheme [16]. A P2P trading mechanism was proposed, in which the decision-making process is modeled by game theory and Shapley value. The game theory approach provides a distributed energy management solution for individual decision-making with respect to the optimality and fairness among consumers. Compared with the existing trading mechanism, the Shapley value trading mechanism helped P2P energy trading achieve better optimization and fairness [17]. A synchronous game theory based on P2P energy trading in a day-ahead market was presented, which will empower participants, without limitation to its number, to decide the transaction price, and VaR is introduced, a risk analysis tool to reduce the risk of transaction failure, maximize the market interests, and increase the success rate of energy trading [18]. Chen K et al. regarded the continuous double auction (CDA) market as a promising mechanism for a P2P market that empowers interactions among prosumers and consumers in distribution grids. For prosumers, achieving optimal operations and maximize profits necessitates acting as price makers and simultaneously optimizing their operations and trading strategies [19].

Existing pieces of literature on P2P energy trading have contributed to the applications of technology and transaction price mechanism modeling. However, there is an imperative gap in user-centered P2P energy trading. Therefore, a P2P energy trading mechanism based on the coalition game theory was devised to bridge the gap. The focus of the coalition game is how to motivate independent decision makers (users) to act together in alignment to elevate their status (or utility) in the game. The coalition's stability lies in two bases: (1) The revenue of the grand coalition surpasses the sum of the benefits that each member would achieve if they were allowed to self-operate. (2) The revenue allocation of the grand coalition should be based on Pareto improved property, which means each member will receive a revenue no less than the benefits under independent action. Among the members, the transparent sharing of information between each other and the enforcement of signed agreements are the keys to maintain the stability of the coalition.

The main contribution of this paper lies in the following aspects:

(1) A peer-to-peer (P2P) energy trading mechanism in the microgrid with distributed photovoltaic distributed generation and battery energy storage systems (BESSs) was devised and users in the microgrids were reasonably classified.

(2) A mixed integer linear programming (MILP) method based on YALMIP [20] (it is a modular language for defining and solving advanced optimization problems, which are written in MATLAB language) is proposed to optimize the decision-making of P2P electricity transactions considering a large number of users with distributed photovoltaic generation and battery energy storage systems (BESSs). The proposed model respects variable real-world constraints, including P2P power trading, microgrid users, customer load profile, PV system, battery energy storage systems (BESSs), and market signals.

(3) Based on the coalition game theory, this paper improves the market intermediate rate price model and introduces the weight variables $\alpha$ and $\beta$. By changing the values of $\alpha$ and $\beta$, the trajectory of P2P transaction price can be obtained, which changes the P2P power transaction price from statics to dynamics, more in line with reality.

The rest of this paper is organized as follows: Section 2 presents the structure of the microgrid P2P energy trading system with distributed photovoltaic generation and battery energy storage systems (BESSs), and proposes a mathematical model. Section 3 describes how to design the P2P power trading mechanism and trading process in this paper. Section 4 discusses and analyzes the results. In Section 5, some concluding remarks are drawn with future research.

## 2. Operational Model Analysis and Mathematical Model Construction of P2P Power Trading

In this section, the structure of the power trading system with PV only based on peer-to-peer (P2P) technology is presented. Among this system, PV, BESSs, microgrid scheduling, and balance and management systems are considered. Furthermore, the mathematical model of each component and the design of the P2P power trading mechanism will be demonstrated.

### 2.1. The Operation Model of P2P Power Trading with PV

In light of the status quo of a P2P in power trading, we propose a P2P power trading system in the microgrid with PV with respect to a single form of energy, i.e., electricity. As shown in Figure 1, energy flow refers to the flow of power between buyers and sellers and cash flow illustrates the cost and income of electricity trade.

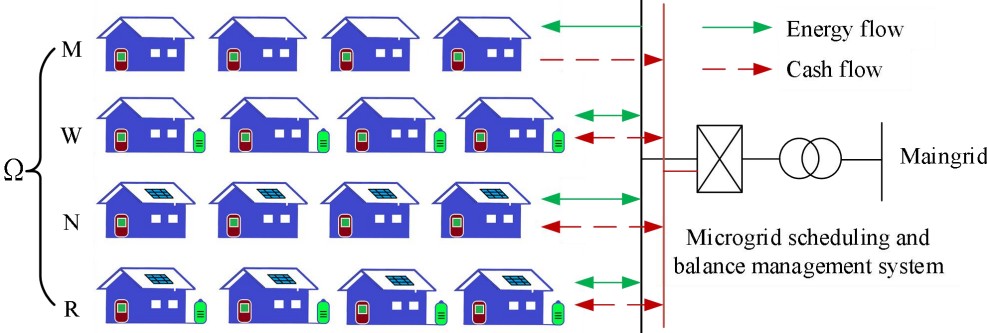

**Figure 1.** Architecture of peer-to-peer (P2P) energy trading system in microgrids.

The main market subjects in the microgrid constructed in this paper mainly include the following types:

Consumers: users without PV can be classified into two major classes: consumers with BESSs (represented by W) and consumers without BESSs (represented by M). By purchasing electricity at a lower price in P2P power trading, consumers can reduce electricity expenditure.

Prosumers: users with PV can be classified into two categories: prosumers with BESSs (represented by N) and prosumers without BESSs (represented by R). Receiving considerable benefit by selling surplus electricity to nearby users or charging energy into storage will motivate users to participate in P2P power trading.

The microgrid operator: a third party is responsible for the operation and maintenance of the microgrid and the settlement of the transaction fees. As with the implementation of the P2P energy system, organizing and supervising the trading platform is the operator's responsibility as well.

In this system, PV powered by the sun can be the energy source of users. Surplus electricity will first be stored in the battery or sold to users in the microgrid, then traded with the grid. On the contrary, purchasing electricity from other microgrid users will be consumers' first choice as well instead of the grid. The P2P energy trading system ensures the users of the microgrid enjoy priority in electricity transactions. The microgrid scheduling and balance management system is operated by the microgrid operator, and what is more, the microgrid operator is also responsible for energy transactions being carried out in an orderly manner, purchase and sale expenses settlement, and interaction with the grid. Its working mechanism is to purchase the surplus electricity of the microgrid users at the P2P transaction price, and then sell it to other microgrid users. If there is surplus electricity, it will sell it to the main grid at the FIT, as opposed to purchasing electricity from the grid at TOU price when the supply of electricity is insufficient. Playing as a third party in the microgrid, the microgrid operator signs a contract with users, offers a real-time monitoring service via a dispatching and balancing management system to execute the contract, and guarantees the priority of the microgrid users in electricity transactions. The microgrid operator functions like an intelligent contract but it is different.

When the microgrid is in short supply, the microgrid operator can make a deal with the grid to satisfy the user's demand. Furthermore, the microgrid operator also provides the users with technical support, such as operation and maintenance service, energy generation prediction, and the like. Therefore, the implementation of the P2P energy transaction in the microgrid can not only increase the effectiveness of energy utilization in the microgrid, but also significantly promote the distributed energy consumption and cut down the electricity costs of the users.

The traditional operation mode of microgrid usually operates in a way that the distributed generation in microgrids is served first to meet the needs of individual customers, and then to trade the surplus electricity, if any, with the grid at the FIT of new energy. Nevertheless, the operation mode of the microgrid integrating peer-to-peer (P2P) technology into power trading will empower individual customers of the microgrid to trade with each other in real time and then sell the surplus power to microgrid at the P2P trading price. The application of P2P technology in energy transactions in microgrids, which enables decentralized transactions, will help with better network and congestion management, allowing a more efficient trading process, assisting with the renewable generation intermittency problem and bringing more social capital into the energy sector.

## 2.2. Mathematical Model

In this paper, a mixed integer linear programming (MILP) model is proposed to optimize operational decisions with respect to a large number of distributed resources and participation in P2P energy trading. In the proposed model, the P2P energy trading mechanism and the real-world limitations on distributed resources, especially PV and BESSs, as well as the fluctuation of market electricity price, is taken into account. The objective function is to minimize the total energy expenditure of all individual customers in the microgrid. In the final step, the model is solved by MATLAB R2016b and CPLEX Optimization Studio v12.8.

### 2.2.1. The Objective Function

This paper assesses the monetary benefit of P2P energy trading in microgrid from the view of the microgrid operator. From this point on, we formulate the objective function by looking for the minimum total energy expenditure of all individual customers in the microgrid.

The inputs for the model include the sets of individual customers and their demand, physical constraints of photovoltaic panels and energy storage, and market price signals. Decision variables include real-world PV generation, actual user electricity consumption, battery charge and discharge, P2P power transaction price, electricity interacting with the main network, and so on.

$$f = \min(C_{btg} + C_{btp} + C_{pv} + C_{ess} - C_{stg} - C_{stp}) \tag{1}$$

where $C_{btg}$ and $C_{btp}$ are the cost of purchasing electricity from the grid and P2P energy market, respectively. $C_{pv}$ and $C_{ess}$ represent the cost of maintaining PV and BESSs, respectively. $C_{stg}$ represents the benefit to the grid from selling electricity and $C_{stp}$ is the benefit to individual customers from selling electricity in the P2P energy market.

(1)　The cost and benefit of interacting with the main grid

$$C_{btg} = P_{sell}^{g}\left(\sum_{i\in\Omega}\sum_{t=1}^{T}Q_{gtu,i,t} + \sum_{i\in W\cup R}\sum_{t=1}^{T}Q_{gtb,i,t}\right) \tag{2}$$

$$C_{stg} = P_{buy}^{g}\left(\sum_{i\in N\cup R}\sum_{t=1}^{T}Q_{pvtg,i,t} + \sum_{i\in W\cup R}\sum_{t=1}^{T}Q_{btg,i,t}\right) \tag{3}$$

where $P_{sell}^{g}$ and $P_{buy}^{g}$ are the selling and purchasing price of electricity of the grid. $Q_{gtu,i,t}$ and $Q_{gtb,i,t}$ are energy bought from the grid for household $\Omega$ and BESSs. $Q_{pvtg,i,t}$ represents the energy sold to the grid from PV and $Q_{btg,i,t}$ is the energy sold to the grid from BESSs discharge.

(2)    The cost and benefit of individual customers participating in P2P energy trading

$$C_{btp} = P_{buy}^{P2P}\left( \sum_{i \in N \cup R} \sum_{t=1}^{T} Q_{pvtm,i,t} + \sum_{i \in W \cup R} \sum_{t=1}^{T} Q_{btm,i,t} \right) \tag{4}$$

where $P_{buy}^{P2P}$ represents the purchasing price of electricity in the microgrid. $Q_{pvtm,i,t}$ is the energy sold to the microgrid generated by PV and $Q_{btm,i,t}$ is the energy sold to the microgrid discharged from BESSs.

$$C_{stp} = P_{sell}^{P2P}\left( \sum_{i \in \Omega} \sum_{t=1}^{T} Q_{mtu,i,t} + \sum_{i \in W \cup R} \sum_{t=1}^{T} Q_{mtb,i,t} \right) \tag{5}$$

where $P_{sell}^{P2P}$ represents the selling price of electricity in the P2P energy trading market. $Q_{mtu,i,t}$ and $Q_{mtb,i,t}$ are the energy bought from the P2P energy trading market by individual customers for daily life and BESSs, respectively

(1)    The cost of maintaining PV

$$C_{PV} = R_{PV}\left( \sum_{i \in N \cup R} \sum_{t=1}^{T} (Q_{pvtg,i,t} + Q_{pvtu,i,t} + Q_{pvtm,i,t}) + \sum_{i \in R} \sum_{t=1}^{T} Q_{pvtb,i,t} \right) \tag{6}$$

where $R_{PV}$ is the operation and maintenance cost of PV per unit electricity. $Q_{pvtu,i,t}$ and $Q_{pvtb,i,t}$ represent the energy generated by PV for household and BESSs.

(2)    The cost of maintaining BESSs

$$C_{ess} = R_{ess}\left( \sum_{i \in R} \sum_{t=1}^{T} Q_{pvtb,i,t} + \sum_{i \in W \cup R} \sum_{t=1}^{T} (Q_{gtb,i,t} + Q_{mtb,i,t} + Q_{btg,i,t} + Q_{btm,i,t} + Q_{btu,i,t}) \right) \tag{7}$$

where $R_{ess}$ is the operation and maintenance cost of BESSs per unit of electricity.

### 2.2.2. Demand Constraint

Depending on the type of individual customers, constraints of customers' demand can be classified into four categories.

(1)    Neither PV nor BESSs

$$D_{i,t} = Q_{mtu,i,t} + Q_{gtu,i,t}. \tag{8}$$

(2)    PV but BESSs

$$D_{i,t} = Q_{mtu,i,t} + Q_{gtu,i,t} + Q_{pvtu,i,t}. \tag{9}$$

(3)    BESSs but PV

$$D_{i,t} = Q_{mtu,i,t} + Q_{gtu,i,t} + Q_{btu,i,t}. \tag{10}$$

(4)    Both PV and BESSs

$$D_{i,t} = Q_{mtu,i,t} + Q_{gtu,i,t} + Q_{pvtu,i,t} + Q_{btu,i,t}. \tag{11}$$

The constraints (8)–(11) ensure that each class of users' needs can be met. Individual customers in electricity deficiency without the distributed generation will first purchase electricity from the P2P energy trading market and then the grid. Households with PV and BESSs will first participate in the P2P market to sell surplus electricity to peers in the microgrid.

### 2.2.3. The Output Constraint of PV

Affected by the solar intensity and temperature, photovoltaic generation is intrinsically intermittent and fluctuant. According to research [21,22], solar intensity is approximately subordinate to the beta distribution during a certain period of time, and the probability density formulation is:

$$f(G) = \frac{\Gamma(\alpha + \beta)}{\Gamma(\alpha)\Gamma(\beta)} \times (\frac{G}{G_m})^{\alpha-1} \times (1 - \frac{G}{G_m})^{\beta-1} \tag{12}$$

where $\Gamma(\bullet)$ is a gamma function in which $G$ and $G_m$ are the real-world and maximum solar intensity during a certain period of time. Shape parameters $\alpha$, $\beta$ can be obtained by making use of the average light intensity and variance during this period of time.

$$\alpha = G_u \times \left[ \frac{G_u \times (1 - G_u)}{\sigma^2} - 1 \right] \tag{13}$$

$$\beta = (1 - G_u) \times \left[ \frac{G_u \times (1 - G_u)}{\sigma^2} - 1 \right] \tag{14}$$

where $G_u$ is the average solar intensity during a certain period of time and $\sigma^2$ represents variance of solar intensity.

Measuring the temperature of the photovoltaic cell directly is not that easy due to the technical limitations, therefore, for the packaged solar cell assembly, the working temperature of the photovoltaic cell can be estimated by measuring the ambient temperature according to function (15).

$$T_w = T_e + 30 \times \frac{G}{1000} \tag{15}$$

where $T_w$ and $T_e$ are the working and ambient temperature of photovoltaic cells, respectively.

The output of the PV can be expressed as follows [23]:

$$P_{pv} = P_s \times \frac{G}{G_s} \times [1 + K(T_w - T_r)] \tag{16}$$

where $G_s$ is the solar intensity under new conditions of the standard test and $G_s = 1000$ W/m$^2$. $P_s$ is the maximum output under the standard test. $K$ is the power temperature coefficient. $T_r$ represents the reference temperature, and $T_r = 25$ °C. As shown in Figure 2, the output of the PV is simulated according to the historical solar intensity and the temperature data.

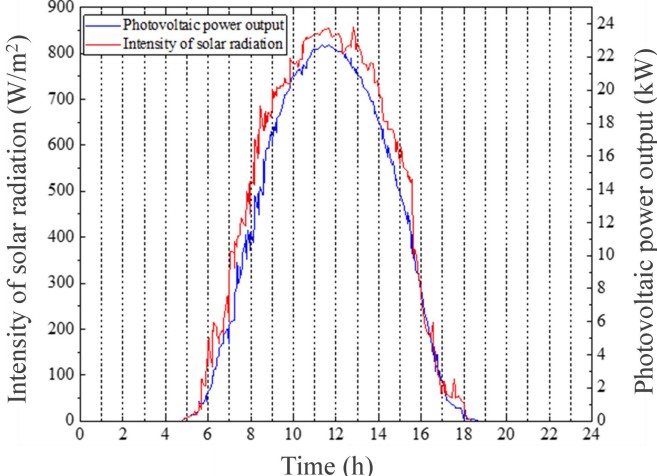

**Figure 2.** Relationship between photovoltaic power output and solar radiation intensity.

Depending on the classes of individual customers, the output of the PV can be classified into two categories.

(1) PV only

$$Q_{pv,i,t} = Q_{pvtu,i,t} + Q_{pvtm,i,t} + Q_{pvtg,i,t}$$
$$Q_{pv,i,t} \leq Q^*_{pv,i,t}$$

(17)

(2) Both PV and BESSs

$$Q_{pv,i,t} = Q_{pvtu,i,t} + Q_{pvtm,i,t} + Q_{pvtg,i,t} + Q_{pvtg,i,t}$$
$$Q_{pv,i,t} \leq Q^*_{pv,i,t}$$

(18)

The electricity generated by the PV can be used for households, BESSs, and trade. The constraints (17) and (18) ensure that the energy generated by all PV systems will not exceed the maximum predicted power generation $Q^*_{pv,i,t}$.

### 2.2.4. The Constraint of Battery Energy Storage Systems

During a period of time, the storage capacity and physical constraints of the BESSs can be modeled by a simplified linear function and described as follows:

$$\begin{cases} Q_{B,i,t} = Q_{B,i,t-\Delta t} \times (1 - \eta_{sd,i,t}) + \left(P_{cha,i,t}\eta_{cha,i,t} - \frac{P_{diss,i,t}}{\eta_{diss,i,t}}\right)\Delta t \\ SOC_{i,t} = \frac{Q_{B,i,t}}{Q_{B,i}} \times 100\% \\ SOC_{\min,i} \leq SOC_{i,t} \leq SOC_{\max,i} \\ 0 \leq P_{cha,i,t} \leq P^{\max}_{cha,i,t}U_{cha,i,t} \\ 0 \leq P_{diss,i,t} \leq P^{\max}_{diss,i,t}U_{diss,i,t} \\ U_{cha,i,t} + U_{diss,i,t} \leq 1 \end{cases}$$

(19)

where $Q_{B,i,t}$ is the storage capacity of BESSs at hour $t$ and $\eta_{sd,i,t}$ represents self-discharge rate. To simplify the computational process, $\eta_{sd,i,t}$ is considered to be [10]. $P_{diss,i,t}$ and $P_{diss,i,t}$ are charging/discharging power. $SOC_{i,t}$ is the storage capacity status of BESSs. $SOC_{\max,i}$ and $SOC_{\min,i}$ are the maximum and minimum of storage capacity status of BESSs related to the property of the BESSs. $U_{cha,i,t}$ and $U_{diss,i,t}$ represent the charge and discharge power state of the BESSs with the opposite binary value of (0 or 1), for the battery cannot be charged or discharged at the same time.

These constraint functions are formulated by considering the state of the battery, the technical limitations on battery capacity, and charging/discharge operation. At a certain moment, the battery can either be charged or discharged, and the maximum storage capacity cannot go beyond its rated capacity. However, this paper highlights the application of P2P in energy trading, so only the basic model of the battery was considered in this paper.

## 3. P2P Energy Trading Mechanism

### 3.1. P2P Energy Trading Process

Figure 3 presents four major stages of the P2P energy trading process in the microgrid containing user demand, initiation of transactions, system verification, and execution of transactions. The transaction process can be implemented based on blockchain technology or other similar distributed networks. The blockchain record transaction information in the form of smart contracts spread to each node through a distributed network and form a consensus [24–26].

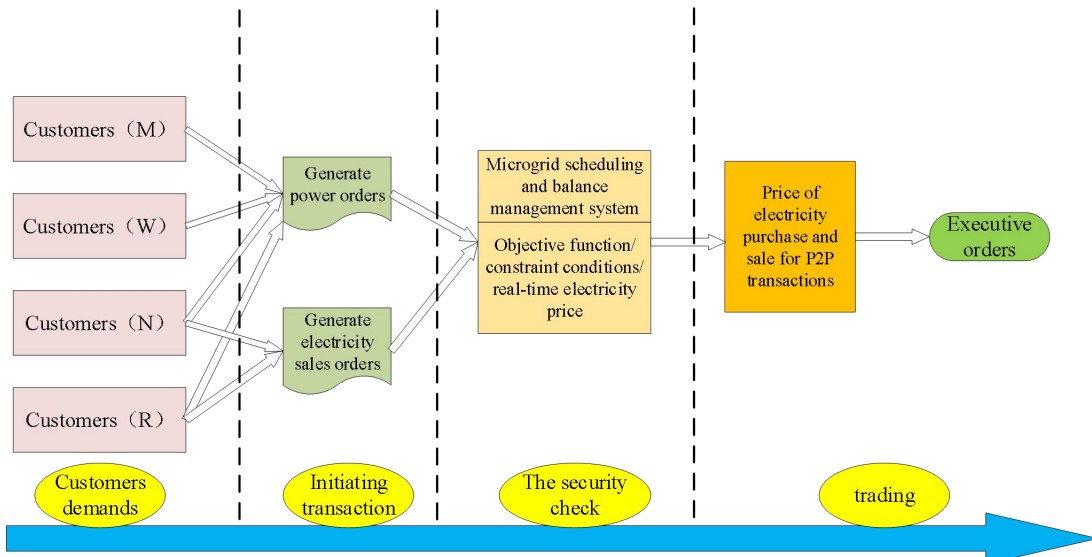

**Figure 3.** P2P energy trading process.

In the stage of the customer demands, users forecast the electricity demand and energy generation at time slot *t* based on reality, and form the demand information.

In the stage of the initiation of the transaction, if the user's net load (load demand minus energy generation) is positive, correspondently, a purchase order will be generated by the P2P energy trading platform, otherwise an electricity sale order will be generated. After that, the orders will be sent to the microgrid through the distributed network.

In the stage of the security check, firstly, the orders will be verified by the microgrid scheduling and balance management system under the constraints of energy balance, the real-time electricity price of the main grid, and the minimum operational cost of the system. Secondly, the optimal P2P purchasing and selling prices will be calculated by MILP optimization, and published to the users.

In the stage of execution of trading, after the orders have been completed, for each participant, the microgrid operator will settle charges of internal orders (orders within the microgrid) at designated P2P purchasing and selling prices and external orders (orders with main grid) at clearing prices.

### 3.2. Coalition Game

The P2P energy trading mechanism is designed to encourage the participation in the decentralized energy market, help with the renewable generation intermittency problem, cut down electricity expenditure, and elevate the monetary benefit of distributed generation owners. The success of P2P trading depends on the design of a reasonable trading mechanism. Many pieces of literature have made relevant studies, such as direct method [27], indirect method [2], and FIT [28]. In direct method, a single distributed resource is controlled and managed by an aggregator, whilst indirect method refers to the strategy in which a central organization sends a signal to the distributed generation owners to affect their electricity consumption and power generation decision. Under the FIT mechanism, however, the engagement of prosumers is more unequivocal for the producers to sell surplus electricity directly to the main grid and purchase electricity from the main grid if needed. We cannot deny the contribution of this research in P2P application, but there is still a key gap in the practical P2P mechanism centered on prosumers.

Therefore, an improved midmarket rate (MMR) model based on the coalition game theory is proposed to bridge the gap between the research and practical application in the P2P energy trading mechanism in this paper. The principle of the MMR [29] is that the reference price of P2P transactions is set as the mean value of the electricity purchasing and selling price of the main grid, and the purchasing and selling price of electricity in the P2P market is based on three scenarios.

(1)　The total energy generation of the prosumers is equal to the overall demand in microgrids. In this case, the P2P trading price equals the average value of the electricity purchasing and selling price of the main grid.

(2)　The total energy generation of the prosumers is greater than the overall demand in microgrids. In this situation, the P2P purchasing price of electricity equals the average value of the electricity purchasing and selling price of the main grid set by the principle of the balance between the cost and the income of the microgrid operator.

(3)　The total energy generation of the prosumers is smaller than overall demand in microgrids, in which case the P2P selling price of electricity is equal to the average value of the electricity purchasing and selling price of the main grid set by the principle of the balance between the cost and the income of the microgrid operator.

The P2P market mechanism mentioned above with little resilience has a limited effect on encouraging participation in P2P energy trading. The focus of a coalition game is how to motivate independent decision makers to act together as a whole to elevate their status (or utility) in the game. Game theory is a mathematical tool used to analyze coping strategies in competitive environments, in which the behavior of one player depends on the behavior of the other players [30]. Game theory can be roughly divided into two categories: noncooperative games and cooperative games. There are two bases for the stability of the coalition: (1) The overall revenue that the grand coalition can achieve is greater than the sum of the revenue when each member operates independently. (2) For each member, the allocation of revenue should abide by the principle of Pareto improvement. In other words, the revenue that each member may attain is no less than that of not joining the alliance. The characteristics of the coalition game ensure those conditions to be satisfied, which means members of the alliance exchange information and the signed agreement must be enforced.

### 3.2.1. The Design of P2P Trading Mechanisms

According to the motivational psychology of users to participate in microgrid P2P trading, the P2P trading prices in microgrids fall in the range of $\left[P_{buy}^g, P_{sell}^g\right]$, which can be converted to linear weighting, and the red line in Figure 4 illustrates the trajectory of P2P trading prices.

$$\begin{cases} P_{P2P} = \alpha P_{buy}^g + \beta P_{sell}^g \\ \alpha + \beta = 1 \end{cases} \tag{20}$$

where $P_{buy}^g$ and $P_{sell}^g$ are the purchasing and selling price of electricity of the main grid. $P_{buy}^{P2P}$ and $P_{sell}^{P2P}$ are the purchasing and selling price of electricity in the P2P market. $Q^n(t)$ is the energy demand of the user $n$ at time slot $t$ and $Q_{pv}^n(t)$ is the energy generated by PV $n$ at time slot $t$.

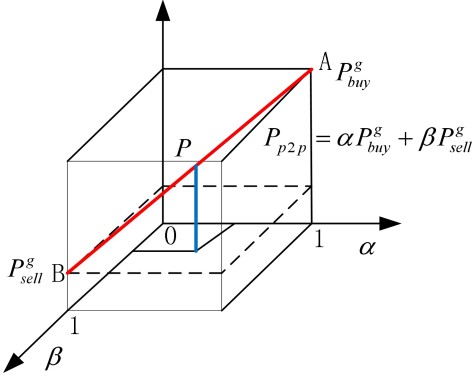

**Figure 4.** The trajectory of P2P trading prices.

The P2P energy trading mechanism is designed to facilitate the consumption of distributed resources in local areas, have revenue of distributed generation retained locally, and give social capital a boost in distributed generation and microgrid.

P2P energy trading price is set in reference to $P_{P2P}$ and the principle of balance between income and cost of microgrid operators. The formula can be expressed as follows:

$$\begin{cases} P_{sell}^{P2P} \times \sum_{n=1}^{N} Q_{pv}^n(t) = P_{buy}^{P2P} \times \sum_{n=1}^{N} Q^n(t) \pm \left| \sum_{n=1}^{N} Q_{pv}^n(t) - \sum_{n=1}^{N} Q^n(t) \right| \times P \\ if \quad \sum_{n=1}^{N} Q_{pv}^n(t) - \sum_{n=1}^{N} Q^n(t) > 0, P = P_{buy}^g \\ if \quad \sum_{n=1}^{N} Q_{pv}^n(t) - \sum_{n=1}^{N} Q^n(t) < 0, P = P_{sell}^g \end{cases} \tag{21}$$

In order to present the pricing process of the P2P market more specifically, we divided the operation of microgrid into three possible scenarios and analyze them respectively.

Scenario 1: production equals demand, as shown in Figure 5.

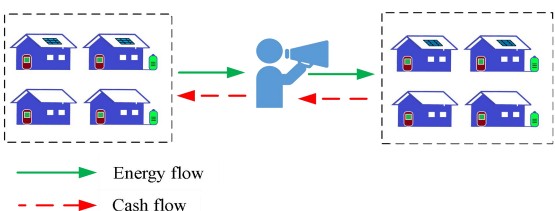

Energy flow
Cash flow

**Figure 5.** Production equals demand.

In this case, P2P trading price can be calculated by $\sum_{n=1}^{N} Q_{pv}^n(t) - \sum_{n=1}^{N} Q^n(t) = 0$, which means

$$P_{sell}^{P2P} \times \sum_{n=1}^{N} Q_{pv}^n(t) = P_{buy}^{P2P} \times \sum_{n=1}^{N} Q^n(t) P_{sell}^{P2P} = P_{buy}^{P2P} = P_{P2P} \tag{22}$$

Scenario 2: production exceeds demand, as shown in Figure 6.

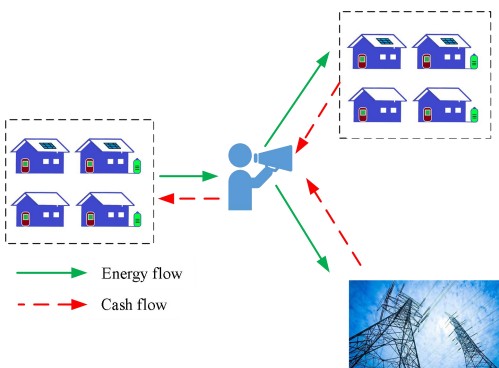

Energy flow
Cash flow

**Figure 6.** Production exceeds demand.

In this case, $\sum_{n=1}^{N} Q_{pv}^n(t) - \sum_{n=1}^{N} Q^n(t) > 0$ and prosumers will be price takers for surplus energy that has to be exported to the grid at $P_{buy}^g$ to maintain the energy balance in the microgrid. Therefore, in the

microgrid, users will be the price taker, which means the P2P energy purchasing price $P_{buy}^{P2P} = P_{P2P}$ and $P_{sell}^{P2P}$ is set based on the equal value of cost and benefit.

$$P_{sell}^{P2P} = \frac{P_{buy}^{P2P} \times \sum_{n=1}^{N} Q^n(t) + (\sum_{n=1}^{N} Q_{pv}^n(t) - \sum_{n=1}^{N} Q^n(t)) \times P_{buy}^g}{\sum_{n=1}^{N} Q_{pv}^n(t)}. \tag{23}$$

Scenario 3: demand exceeds production, as shown in Figure 7.

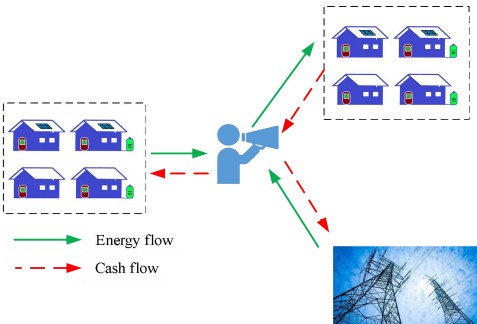

**Figure 7.** Production is less than demand.

In this case, $\sum_{n=1}^{N} Q_{pv}^n(t) - \sum_{n=1}^{N} Q^n(t) < 0$ and users in energy deficiency will be price takers for the necessary amount of energy that has to be imported from the main grid at $P_{sell}^g$ to ensure the demand is met in the microgrid. Therefore, in the microgrid, prosumers will be the price taker, which means the P2P energy selling price $P_{sell}^{P2P} = P_{P2P}$ and $P_{buy}^{P2P}$ is set based on the equal value of cost and benefit.

$$P_{buy}^{P2P} = \frac{P_{sell}^{P2P} \times \sum_{n=1}^{N} Q_{pv}^n(t) + (\sum_{n=1}^{N} Q^n(t) - \sum_{n=1}^{N} Q_{pv}^n(t)) \times P_{sell}^g}{\sum_{n=1}^{N} Q^n(t)}. \tag{24}$$

### 3.2.2. Assumptions

Based on the coalition game theory, we make three assumptions for the P2P trading mechanism:

(1)　All participants are rational decision makers.
(2)　All participants must comply with the internal agreement and forbid members to withdraw casually from the grant union.
(3)　All participants enjoy priority in purchasing energy from the microgrid and selling electricity to peers in the microgrid.

In Figure 1, prosumers will sell surplus electricity to the microgrid operator and customers will buy energy from the microgrid operator. According to the real-time monitoring data of the microgrid scheduling and balance management system, the microgrid operator will balance the internal system by exporting surplus electricity to the main grid or importing deficient energy from the main grid. The P2P trading price is set based on the principle that cost is equal to revenue. Meanwhile, the main monetary benefit of microgrid operator comes from registration fees for users and services charges, including PV maintenance, PV output prediction, etc.

## 4. Case Study

To show the feasibility of the proposed P2P energy trading system, we tested it on an artificial situation, which contained 30 households, including 5 residential customers without PV and BESSs, 5 with BESSs only, 10 with PV only, and 10 with both PV and BESSs. For each residential customer in the microgrid, it was assumed that their maximum demand is 80 kW. Gross capacity of solar panels and BESSs whose specification is the same as PV were assumed to be 115 kW and 50 kW, respectively. The real-time energy usage of residents can be demonstrated in the scheduling and balance management system of microgrid. The value of the FIT price was assumed to be 0.381 ¥/kWh. The tiered selling price of electricity from the grid to residential is shown in Figure 8.

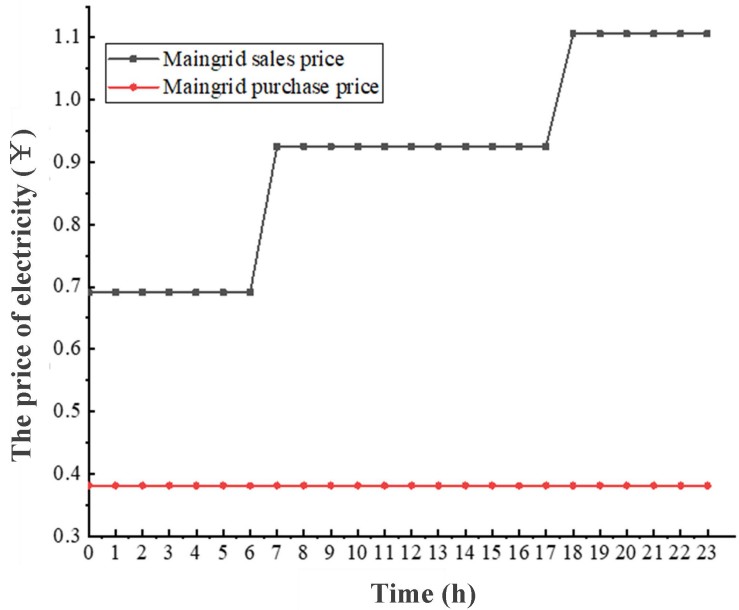

**Figure 8.** Main grid real-time purchase and sale price.

In this paper, the mixed integer linear programming (MILP) method based on YALMIP was used to solve the model. YALMIP is a solution toolbox written in MATLAB language. It can optimize the objective function by calling on a variety of commercial optimization solvers, such as LPSOLVE, CPLEX, and GRUOBI. In this article, we wrote the program code through the YALMIP toolbox and called CPLEX in the MATLAB operation environment to optimize the solution. Finally we got the optimal price of the 24 h P2P purchase and sale price in the microgrid, as shown in Figure 9a,b, and the minimum total operating net cost of the microgrid fluctuated with the value of parameter $\alpha$ and $\beta$, as shown in Figure 10.

In Figure 9a, the line of the main grid sales price is always at the top, indicating that purchasing electricity from the P2P market is always cheaper than that in the main grid no matter how the value of $\alpha$ and $\beta$ change, which is conducive to encouraging microgrid users to participate in P2P transactions. In Figure 9b, the FIT price is fixed and the P2P selling price of electricity fluctuates with the value of $\alpha$ and $\beta$. When $\alpha = 0.6$, $\beta = 0.4$, the lower limit of the optimal P2P selling price of electricity obtained for the P2P selling price of electricity is higher than the FIT price at this point, and it keeps rising. Therefore, when $\alpha = 0.6$, $\beta = 0.4$, the minimum operating net cost of the microgrid can be achieved. In Figure 10, with the change of $\alpha$ and $\beta$, the net operating cost of microgrid gradually increases, which is due to the increase in the P2P transaction price, and the growth rate of internal cost of the microgrid is greater than that of internal revenue. Simultaneously meeting the conditions of the minimum operating net cost of the microgrid and a higher P2P sales price than FIT, we thus got the best P2P energy transaction price at $\alpha = 0.6$, $\beta = 0.4$.

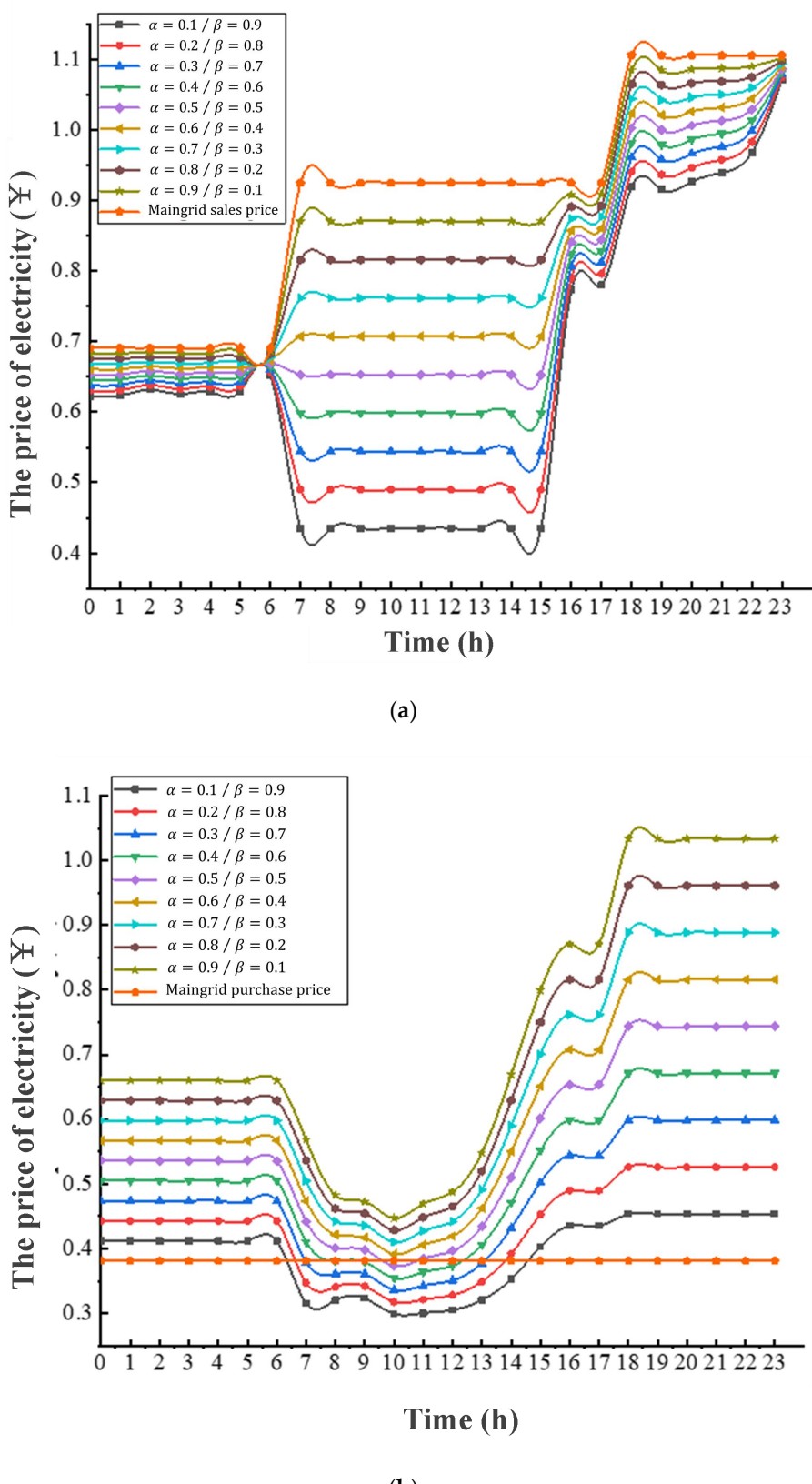

**Figure 9.** P2P energy transaction price. (**a**) Energy purchase price of P2P transaction. (**b**) Energy sale price of P2P transaction.

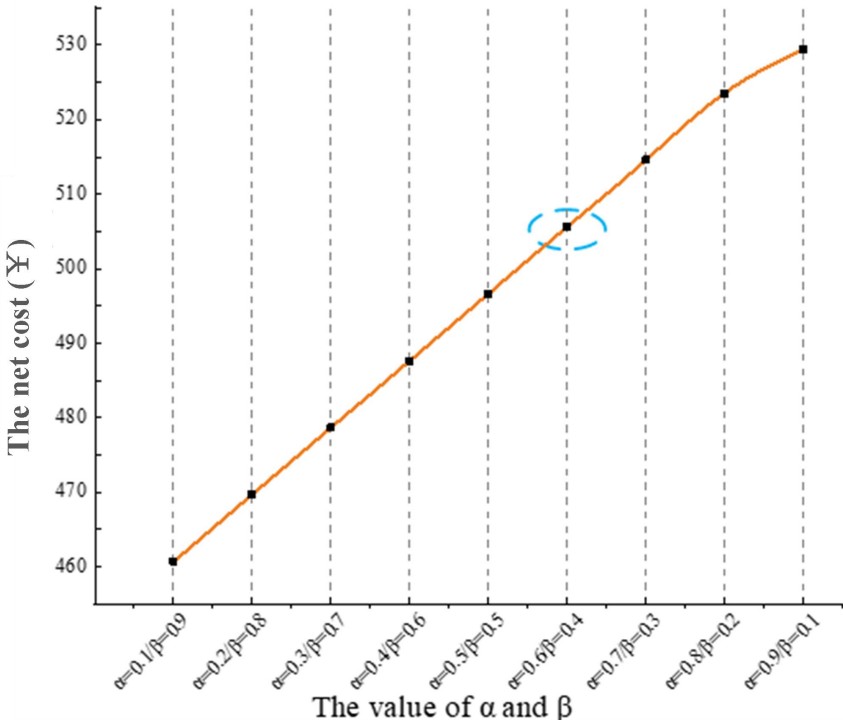

**Figure 10.** Net cost of 24 h operation of microgrid.

### 4.1. Analysis of the Overall Operation of the Microgrid

As shown in Figure 11, from 0:00 to 6:00, PV stops functioning and households can meet the demand by discharging the battery and importing electricity from the main grid. In the day, from 7:00 to 16:00, the PV starts to generate electricity, the outputs of which can satisfy the need for overall demand in the microgrid. To be exact, households with PV and BESSs will first meet their electricity demand, and then charge the battery to get revenue by selling surplus electricity to peers in the microgrid or exporting to the main grid. From 17:00 to 23:00, we can see that PV systems do not generate electricity and the energy resource for households comes from BESSs discharging or importing from the main grid. Figure 11 illustrates that energy transactions in the P2P market mainly occur from 7:00 to 16:00 of the day, of which the green part of the bar chart stands for energy purchased from the P2P market by households and the orange for energy sold in the P2P market by households. If PV outputs exceed the demand of prosumers, it will be more attractive for prosumers to first trade electricity in the P2P market, and then export surplus energy to the main grid cleared at retail price under the organization of the P2P operator, as the selling price of energy in the P2P market is relatively higher than that in the main grid and the purchasing price is quite the opposite. In general, the proposed P2P energy trading mechanism can achieve favorable savings for households with PV systems installed, decrease the electricity costs for users in the microgrid, and realize a win–win situation for prosumers and customers.

### 4.2. Operational Performance of Prosumers Analysis

The operational performance of prosumers with PV generation and BESSs is illustrated in Figure 12. The PV outputs from 7:00 to 17:00 can be divided into four categories according to the method of energy usage: (1) Self-consumption for prosumers. (2) Charging the battery. (3) Trade in P2P market. (4) Export to the main grid. P2P energy trading for the distributed generation is a promising mechanism that enables the surplus energy of prosumers to be consumed in the local area, giving rise to the revenue of prosumers, helping with electricity cost reduction. As the Chinese government gradually reduces the subsidy for the new energy feed-in tariffs, the investment of social capital in the development and construction of new energy is also gradually reducing, which to some extent hinders the development

of new energy industry. Therefore, a scientific and reasonable P2P trading mode is conducive to encouraging social capital to invest in new energy construction.

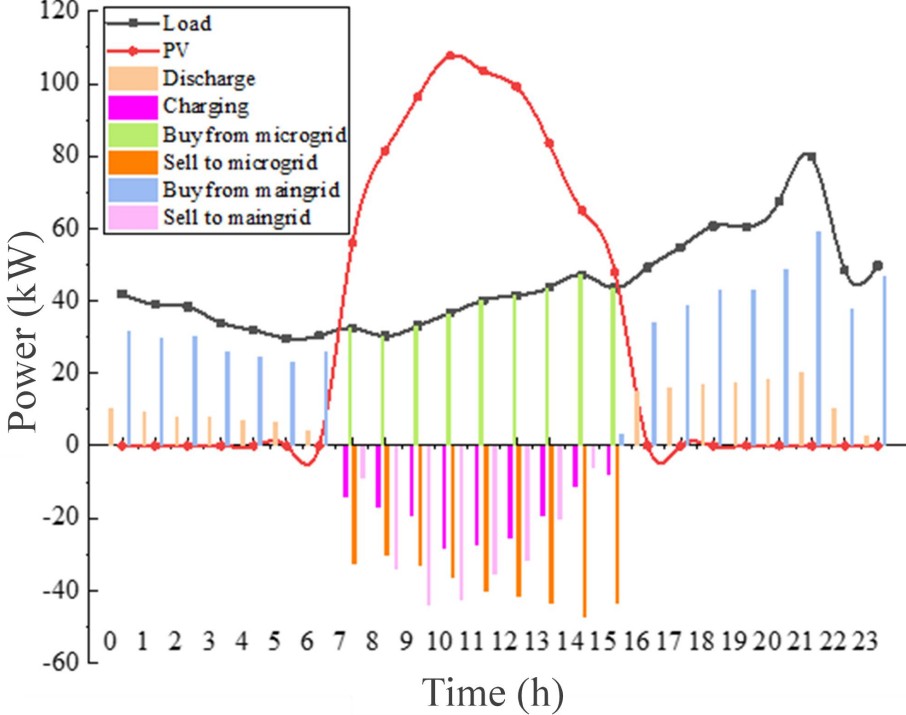

**Figure 11.** A 24 h operation state of the microgrid.

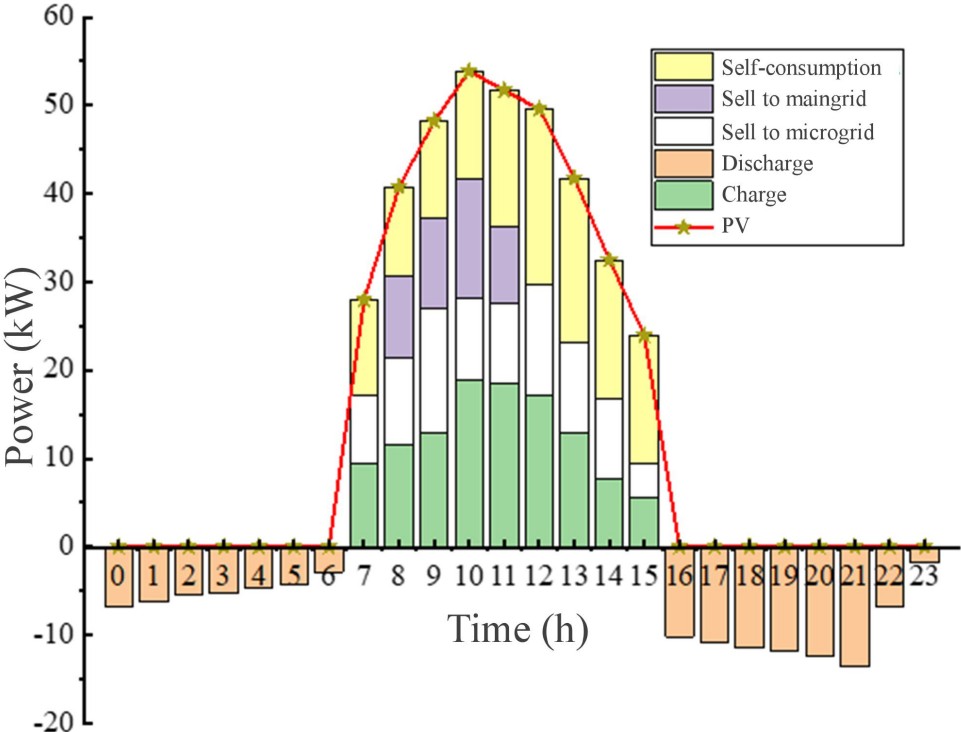

**Figure 12.** Application classification of photovoltaic power generation.

For microgrid users with PV only, the actual operation is similar to that of microgrid users with PV and BESSs. The power generated by PV production first meets its own needs, and then is sold to

the P2P market to satisfy the demand of peers in the microgrid. If there is any surplus power, it will be sold to the main grid through the microgrid operator. The difference between the households with PV and battery is that they do not need to charge the battery. When the electricity produced by PV cannot meet their own needs, the only way to obtain electricity is to purchase electricity from the P2P market or the main grid, while the households with battery can meet their own needs by discharging the battery to reduce electricity expenditure.

In Figure 13, the energy sources of households with PV and BESSs is depicted. From 0:00–6:00 and 16:00–23:00, PV stops functioning and households prefer to get electricity from battery discharge. If this is not enough, households will import electricity from the main grid under the help of microgrid operator. From 7:00–15:00, we can see that households like to treat PV as the first energy resource, as PV starts producing energy at this period. Owing to the friendly mode of electricity usage containing PV and BESSs, the PV output abandonment rate is considerably reduced.

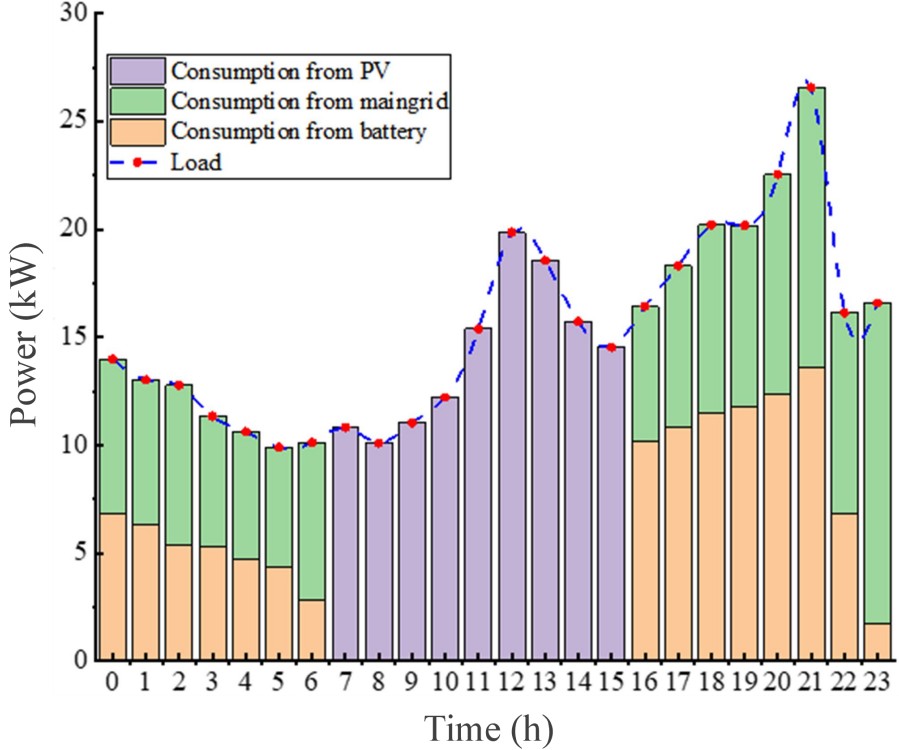

**Figure 13.** Sources of power consumption of consumers with PV and battery.

*4.3. Operational Performance of Households Only with BESSs Analysis*

Figure 14 demonstrates the energy consumption profile of households only with BESSs throughout the day. We can see that from 0:00–7:00 the primary energy resources of households are battery discharge and the main grid. During the day, from 8:00–16:00, households purchase electricity from the P2P market for self-consumption and battery charge. Due to fact that the P2P price is more favorable than the retail price of the main grid, households prefer to purchase electricity form the P2P market rather than the main grid. From 17:00–23:00, the P2P trading market is off because PV outputs are not available and the energy consumed by households comes from battery discharge and the main grid.

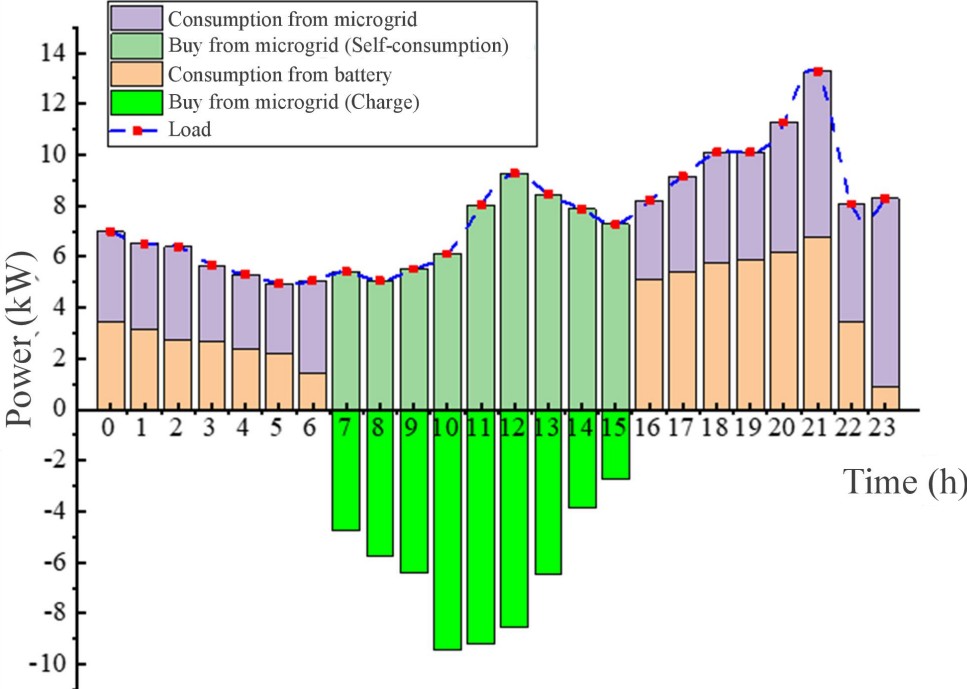

**Figure 14.** Sources of power consumption of consumers with battery.

## 5. Conclusions

In this paper, a P2P energy trading model in microgrids with PV and BESSs is proposed. Households could broadly be divided into four major classes. Among these, households with PV could share surplus electricity with other prosumers. Based on the coalition game theory, the trading price is set by considering the minimum overall energy consumption in microgrids under various scenarios. The model is illustrated in a simulation framework. As a result, this mechanism for P2P energy trading promises a lot for changing households' preference toward their own generation and self-consumption, reducing the dependence of the distributed generation on subsidies, encouraging the scale of prosumers both in the business and the residential sectors, assisting with the distributed and renewable generation intermittency problem, and enabling more efficient and effective network monitoring.

Looking at economic aspects, with the help of the P2P energy trading model, it will lower households' electricity expenditure and bring considerable income for prosumers. If the household is in electricity deficiency, the extra energy generated by PV and stored in the battery will be available, which will cut down the cost of purchasing electricity. The BESSs participating in P2P energy markets at peak hours will unlock more benefits for the owners of BESSs.

As for social benefits, P2P energy trading for the distributed generation is a mechanism that shifts the energy, delivering energy in a more flexible and decentralized way, enabling the surplus energy of prosumers to be consumed in the local area. It will bring more social capital to the energy sector, reduce state funds in power infrastructures, and facilitate the development of distributed and renewable energy. Hence, P2P energy trading will lead households to shift from consumers to prosumers. Correspondingly, it will increase the need for equipment for distributed energy generation and BESSs, assist with redundant capacity elimination, and indirectly give rise to the employment demand.

Note that some shortcomings in this paper are that some factors, including the environmental benefits, photovoltaic distributed construction costs, and investment recovery cycle, are not considered. From this point on, factors mentioned above will be taken into account in future research to elevate existing P2P energy trading models and to inspire the application of P2P technology in the energy sector.

**Author Contributions:** J.D. and H.H. supervised the proposed research work, whereas S.N. conceptualized the proposed research work, collected the data, programmed and analyzed the methodology, and wrote the original draft, J.L. and Y.W. helped in review writing and editing with validation of research work. All authors have read and agreed to the published version of the manuscript.

**Funding:** This research was funded by the Beijing Social Science Foundation Research Base Project (18JDGLB037).

**Conflicts of Interest:** The authors declare no conflict of interest.

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
