# Peer review of "Optimization of Peer-to-Peer Power Trading in a Microgrid with Distributed PV and Battery Energy Storage Systems"

_sustainability, doi:10.3390/su12030923_

Round 1

Reviewer 1 Report

The subject of this article is within the scope of this journal. Overall, the technical (mathematical) portion of the work and the combination of the employed methods seem sound. However, the exposition is not acceptable. Significant and major revisions are required to improve the quality of the work for publication. The text is occasionally hard to read and the article needs significant improvements in English grammar, punctuation and structure.

This article presents a p2p power trading method for distributed consumers that may have PV and battery energy storage systems (BESSs). Authors first provide an introduction and then a literature review of the recent work in p2p trading. For the technical section, an optimization problem is formed to minimize the electricity cost from the point of view of the uG operator.

The technical section of the work seems sound. However, numerous typing and grammatical errors reduce the quality of this work. Here are few instances that caught my eye: 

- From a reader’s point of view, the title may first appear to be for a review paper. However, a major portion of this work is dedicated to technical work.

- There are many typing errors in this article. For example, Typos in line 13.

- It is common practice to mention all Figures in the body of the text. 

- Many in-line equations are not aligned; e.g. line 249.

- Nomenclature section or additional references are occasionally needed (e.g. line 119): it is common practice to provide details for YALMIP when first mentioned.

- Fig. 1 is not consistent with the text in line 145. (M is classified as “with ES”)

Author Response

Point 1: From a reader’s point of view, the title may first appear to be for a review paper. However, a major portion of this work is dedicated to technical work. 

Response 1: Thank you for your advice ,we have changed the title of the article to “Optimization of Peer-to-Peer power trading in microgrid with distributed PV and battery energy storage systems”

Point 2: There are many typing errors in this article. For example, Typos in line 13.

Response 2: Thank you for your advice, the spelling of the word has been corrected

Point 3: It is common practice to mention all Figures in the body of the text.

Response 3: Thank you for your advice. After modification, all Figures are mentioned in body of the text.

Point 4: Many in-line equations are not aligned; e.g. line 249.

Response 4: Thank you for your advice. All in-line equations in body of the text have been modified according to the paper template

Point 5: Nomenclature section or additional references are occasionally needed (e.g. line 119): it is common practice to provide details for YALMIP when first mentioned.

Response 5: Thank you for your advice. We have described YALMIP in detail and cited the relevant reference [20] as supporting evidence

Point 6: Fig. 1 is not consistent with the text in line 145. (M is classified as “with ES”)

Response 6: Thank you for your advice. We have modified the type of customer represented by the letter M in figure 1 and made corrections in the text

Reviewer 2 Report

The paper is devoted to a very interesting topic. This paper focuses on a peer to peer energy trading model in microgrid including PV and battery energy storage systems. 

Adequate mathematical models are applied. Results are well discussed and the reference list is adequate. The paper is made qualitatively. Therefore, the manuscript is recommended for publication.

Author Response

Thank you for your advice.

Reviewer 3 Report

The paper “Research on Peer-to-Peer power trading in microgrid with distributed PV and battery storage”, provides a detailed introduction that summarizes the current situation of the P2P of energy trade. For this, the authors are based on an extensive and current bibliography.

The structure of the paper is clear and the authors analyze various situations using the MILP and Matlab simulation. An interesting mathematical model has been developed. This model covers the variables and restrictions of the system under study.

The methodology used by the authors is adequately explained, the results are clearly presented and the conclusions are supported by them.

I consider it is a very interesting work that addresses a current problem and, in my opinion, the paper can be published with the following minor revisions:

Throughout the document, the expressions embedded in the text as equations are displaced with respect to it, see as an example lines 203, 204, 205 ... Figure 2, the power unit in the legend must be kW instead of KW. Equation 19, separate the lines for clarity. Subscripts and superscripts of different expressions overlap. Lines 392 and 394, the power unit must be kW instead of KW. Figures 11, 12 and 13, the power unit must be kW instead of kw Figure 11, change “power” to “Power”.

Author Response

Point 1: Throughout the document, the expressions embedded in the text as equations are displaced with respect to it, see as an example lines 203, 204, 205 ... 

Response 1: Thank you for your advice. We have revised the format of the expressions of equations embedded in the text.

Point 2: Figure 2, the power unit in the legend must be kW instead of KW.

Response 2: Thank you for your advice. We have changed the power unit KW to kW in Figure 2

Point 3: Equation 19, separate the lines for clarity. Subscripts and superscripts of different expressions overlap.

Response 3: Thank you for your advice. We have modified Equation 19 as required

Point 4: Lines 392 and 394, the power unit must be kW instead of KW

Response 4: Thank you for your advice. We have changed all the units of power in this article from KW to kW

Point 5: Figures 11, 12 and 13, the power unit must be kW instead of kw.

Response 5: Thank you for your advice. We have changed the power unit of KW to kW in all the figures in the full text

Point 6: Figure 11, change “power” to “Power”.

Response 6: Thank you for your advice. We have changed the “power” to “Power” in Figure 11

Reviewer 4 Report

The line 13 has photovoltaicdistributed Word, and may be photovoltaic distributed 

Figure 2 . Has "Photovoltaic power output (KW)" and the correct unit is "kW"  

From line 392 to 394 there is the same error 80 kW 115 kW and 50 kW

Figure 11,12 and 14. Has de reverse problema. "Power (kw)" and the correct form "Power (kW)"   

Figure 13. Has  Power (kw) and the correct form is "Power (kW)" and the label "Laod" may be "Load"

Author Response

Point 1: The line 13 has photovoltaicdistributed Word, and may be photovoltaic distributed 

Response 1: Thank you for your advice. We have “photovoltaicdistributed” changes for “photovoltaic distributed”

Point 2: Figure 2 .Has "Photovoltaic power output (KW)" and the correct unit is "kW"

Response 2: Thank you for your advice. We have changed the power unit of KW to kW in all the figures in the full text

Point 3: From line 392 to 394 there is the same error 80 kW 115 kW and 50 kW.

Response 3: Thank you for your advice. We have changed all the units of power in this article from KW to kW

Point 4: Figure 11,12 and 14. Has de reverse problema. "Power (kw)" and the correct form "Power (kW)"

Response 4: Thank you for your advice. We have changed the power unit of KW to kW in all the figures in the full text

Point 5: Figure 13. Has Power (kw) and the correct form is "Power (kW)" and the label "Laod" may be "Load.

Response 5: Thank you for your advice. We have changed the power unit of KW to kW, and the label “Laod” changed to “Load” in figure13
